# Daily Water Mapping and Spatiotemporal Dynamics Analysis over the Tibetan Plateau

Qi Feng [1], Kai Yu [2,3] and Luyan Ji [2,3,4,*]

1 Institute of Remote Sensing Application in Public Security, People's Public Security University of China, Beijing 100038, China

2 Aerospace Information Research Institute, Chinese Academy of Sciences, Beijing 100094, China;

3 Key Laboratory of Target Cognition and Application Technology, Chinese Academy of Sciences, Beijing 100190, China

4 School of Electronic, Electrical and Communication Engineering, University of Chinese Academy of Sciences, Beijing 100049, China

* Correspondence: jily@mail.ustc.edu.cn

**Abstract**

The Tibetan Plateau, known as the "Asian Water Tower", contains thousands of lakes that are sensitive to climate variability and human activities. To investigate their long-term and short-term dynamics, we developed a daily surface-water mapping dataset covering the period from 2000 to 2024 based on MODIS daily reflectance time series (MOD09GQ/MYD09GQ and MOD09GA/MYD09GA). A hybrid methodology combining per-pixel spectral indices, superpixel segmentation, and fusion of Terra and Aqua results was applied, followed by temporal interpolation to produce cloud-free daily water maps. Validation against Landsat classifications and the 30 m global water dataset indicates an overall accuracy of 96.89% and a mean relative error below 9.1%, confirming the robustness of our dataset. Based on this dataset, we analyzed the spatiotemporal evolution of 1293 lakes (no less than 5 km$^2$). Results show that approximately 87.7% of lakes expanded, with the fastest growth reaching +43.18 km$^2$/y, whereas 12.3% shrank, with the largest decrease being −5.91 km$^2$/y. Seasonal patterns reveal that most lakes reach maximum extent in October and minimum extent in January. This study provides a long-term, cloud-free daily water mapping product for the Tibetan Plateau, which can serve as a valuable resource for future research on regional hydrology, ecosystem vulnerability, and climate–water interactions in high-altitude regions.

**Keywords:** Tibetan Plateau; water mapping; daily; MODIS

## 1. Introduction

The Tibetan Plateau has numerous lakes and is the source of many major rivers, and is widely known as the "Asian Water Tower". These lakes are highly sensitive to environmental changes and play a critical role in regional hydrological and ecological processes. In recent years, due to global warming, precipitation in the Tibetan Plateau has shown an increasing trend, leading to a significant increase in both the number and area of lakes in this region [1–5]. Understanding these dynamics is essential for evaluating water resource stress [6], ecosystem stability [7], hydrological processes modeling [8], and downstream impacts on nearly 22% of the global population (about 2 billion people).

Currently, these studies have mainly focused on the long-term trends of water body changes (typically at an annual scale), in order to investigate the driving factors behind

them. Yet, in addition to long-term changes, water bodies on the Tibetan Plateau also exhibit significant intra-annual variations, which mainly include the following two aspects. (1) The first is water expansion and shrinkage caused by precipitation, evaporation, etc. Short-duration heavy rainfall can rapidly enlarge lake areas. For example, the two heavy rainfall events prior to 22 August 2011, along with the subsequent continuous precipitation, were the main causes of the large-scale outflow of water from Zhuonai Lake [9]. (2) The second is the freezing and thawing of water due to temperature fluctuations [10]. These short-term changes are also influenced by climatic factors and can indirectly reflect climate variability. Therefore, to effectively capture the dynamic characteristics of water bodies, high-frequency monitoring is required [11,12] in this region.

Major water-mapping products covering the entire Tibetan Plateau based on remote sensing satellite data are listed in Table 1. We can see that the products are mostly based on publicly available satellite data such as Landsat and Moderate Resolution Imaging Spectro-radiometer (MODIS). As a result, their temporal and spatial resolutions are constrained by the remote sensing data used. While Landsat data has a higher spatial resolution (30 m) but a relatively longer revisit cycle (16 days), water mapping products based on Landsat typically have a high spatial resolution, but are usually generated on an annual or, at best, monthly basis (such as products No. 1, 6, 8, 10, 14, and 17). Conversely, MODIS data has lower spatial resolution but provides daily global coverage; thus, water mapping using MODIS reflectance products generally features lower spatial resolutions (250 m or 500 m) but relatively higher temporal resolutions (8 days or daily), such as products No. 15, 16, and 18. Moreover, most existing datasets neglect lake ice phenology, despite its importance as an indicator of climate change on the Plateau.

Due to the inherent trade-off between spatial resolution and revisit frequency in satellite sensor design, a single sensor cannot simultaneously provide both high temporal and high spatial resolution data [13]. To effectively monitor the highly dynamic nature of water, daily global data coverage is essential, making MODIS a suitable choice. On the Tibetan Plateau, lakes larger than 10 km$^2$ account for approximately 90% of the total lake area [14], suggesting that moderate spatial resolution can still meet the requirements for large-scale water mapping. Therefore, we still employ the daily reflectance products from MODIS in this study for water mapping.

Currently, existing products rarely exploit the full suite of MODIS observations (Terra and Aqua; reflectance and land surface temperature), leaving challenges in both classification accuracy or cloud contamination unaddressed. Klein et al. [15–17] generated a global daily water mapping product using the 250 m MODIS daily surface reflectance products, MOD09GQ and MYD09GQ, named Global WaterPack. By combining data from both the Terra and Aqua satellites, their approach significantly reduced data gaps caused by clouds and missing observations. However, their product relied solely on MOD09GQ and MYD09GQ time series, which contain only two spectral bands: the red and near infrared (NIR) band. Yet, compared to red and NIR bands, the shortwave infrared (SWIR) region is more crucial for water identification due to the strong absorption in this region [18]. As a result, using SWIR bands can help to distinguish water from other land cover types. Moreover, their products did not account for lake ice cover, which is a key indicator of climate change impacts on the Tibetan Plateau. Monitoring ice phenology requires daily mapping results, as ice formation and melting are highly dynamic processes sensitive to temperature variations. Ji et al. [19] extracted global daily water bodies using MOD09GA, which provides a broader range of spectral bands, and also includes the detection of lake ice cover. However, MOD09GA has a coarser spatial resolution of 500 m. And only the data from MODIS Terra satellite is utilized. As a result, its spatial resolution is lower than that of

the Global WaterPack, and its performance in areas with persistent cloud cover is also less reliable due to the lack of complementary observations from the MODIS Aqua satellite.

The Tibetan Plateau is characterized by a high frequency of cloud cover, particularly during the summer months [20]. To improve the accuracy of water body mapping under frequent cloud cover, this study integrates all available MODIS daily surface reflectance products from both Terra and Aqua satellites, including MOD09GQ/MYD09GQ and MOD09GA/MYD09GA. A hybrid framework is designed, incorporating per-pixel spectral analysis, superpixel segmentation using K-means-based Connection Center Evolution (KCCE) [21], multi-sensor fusion, and temporal gap-filling. The 09GQ series provides higher spatial resolution (250 m), while the 09GA series offers more spectral bands, which enhances classification accuracy. By combining data from both Terra and Aqua satellites, the negative impacts of cloud contamination over the Tibetan Plateau can be effectively mitigated. In addition, to obtain more accurate information on lake ice, we further incorporated the MODIS Land Surface Temperature (LST) product (MOD11A1) to support and enhance the detection of lake ice phenology. Based on the derived daily water body dataset, we further conduct a comprehensive spatiotemporal analysis of lake dynamics across the Tibetan Plateau over the past 25 years, which aimed to capture both long-term interannual trends and intra-annual seasonal variation patterns. The main contributions of this work are:

- It provides the a cloud-free daily lake mapping dataset for the TP with demonstrated high accuracy and consistency.
- It introduces an improved methodological framework that integrates multi-source MODIS data and explicitly accounts for snow/ice cover, addressing limitations in prior products.
- It offers new insights into TP lake dynamics over the past 25 years, highlighting both widespread expansion and localized shrinkage, thereby advancing understanding of hydrological and environmental change in high-altitude regions.

**Table 1.** Description of water mapping products on the Tibetan Plateau.

| Region | No. | Reference | Temporal Range | Temporal Resolution | Spatial Resolution | Data Source |
|---|---|---|---|---|---|---|
| Tibetan Plateau | 1 | [22] | 1970, 1990, 2000, 2010 | Single-temporal (4 periods) | 1970: 80 m; 1990: 28.5 m | Landsat MSS/ETM+, GeoCover circa 1990/2000 |
| | 2 | [14] | 2005–2006 | Single-temporal (1 period) | 2000: 14.25 m; 2010: 30 m | CBERS CCD, Landsat ETM+ |
| | 3 | [23] | 2000–2013 | Monthly (14 years) | 500 m | MODIS (MOD09A1) |
| | 4 | [24] | 1960, 2005, 2014 | Single-temporal (3 periods) | 1960: 1:250,000; 2005: 19.5 m; 2014: 16 m | Historical survey, CBERS CCD, Landsat TM/ETM+/OLI, GF-1 WFV/PMS |
| | 5 | [25] | 2015–2017 | Monthly (3 years) | 40 m | Sentinel-1 SAR |
| | 6 | [4] | 1991–2018 | Yearly (18 years) | 30 m | Landsat TM/ETM+/OLI |

**Table 1.** *Cont.*

| Region | No. | Reference | Temporal Range | Temporal Resolution | Spatial Resolution | Data Source |
|---|---|---|---|---|---|---|
| China | 7 | [26] | 1960–1980 | Single-temporal (1 period) | 1:250,000 | Historical survey |
| | 8 | [27] | 1990–2000 | Single-temporal (2 periods) | 30 m | Landsat TM/ETM+ |
| | 9 | [28] | 2005–2006 | Two-temporal (Wet and dry seasons) | 20–30 m | CBERS CCD, Landsat TM/ETM+ |
| | 10 | [29] | 2005–2008 | Single-temporal (1 period) | 30 m | Landsat TM/ETM+ |
| Global | 11 | [30] | 2004 | Single-temporal (1 period) | 200 m/1000 m/5000 m/25,000 m | MGLD, LRs, WRD, DCW, ArcWorld, WCMC, GLCC |
| | 12 | [31] | 2000–2015 | Yearly (16 years) | 250 m | MODIS (MOD44C) |
| | 13 | [32–34] | 1992–2015 | Monthly (24 years) | 25 km | SSMI, ERS |
| | 14 | [35] | 1984–2015 | Monthly (32 years) | 30 m | Landsat TM/ETM+/OLI |
| | 15 | [15,16] | 2003–2022 | Daily (20 years) | 250 m | MODIS (MOD09GQ, MYD09GQ) |
| | 16 | [19] | 2001–2024 | Daily (25 years) | 500 m | MODIS (MOD09GA) |
| | 17 | [36] | 1992–2018 | Monthly/Bimonthly (27 years, excluding frozen seasons) | 30 m | Landsat TM/ETM+/OLI |
| | 18 | [37] | 2000–2018 | 8-day interval (19 years) | 250 m | MODIS (MOD09Q1) |

## 2. Study Area and Data

### 2.1. Study Area

The Tibetan Plateau , the world's most extensive and highest plateau, is situated in central Asia (25°59′ N–39°49′ N; 73°29′ E–104°40′ E). It has an average elevation exceeding 4000 m above sea level and a total area of approximately $2.5 \times 10^6$ km². As the "water tower of Asia", it supplies fresh water to ∼22% of Earth's population downstream for agricultural irrigation, industrial production, and domestic use [38].

Climatically, the climate of the Tibetan Plateau is mostly influence by the westerlies in winter and the Asian monsoon in summer. This leads to significant spatial and seasonal variations in temperature and precipitation across the plateau. Mean monthly air temperatures, recorded at China Meteorological Administration (CMA) weather stations, range from approximately −10 °C in winter to ∼10 °C in summer [25]. Due to heterogeneous geography, precipitation is highly seasonal and spatially uneven: 60–90% of annual precipitation falls between June and September, and its amount gradually decreases from the southeast to the northwest, with an east-to-west gradient of 335–430 mm/y [4].

### 2.2. Daily Reflectance Time Series

The data sources used in this product are mainly the MODIS daily reflectance time series (https://modis.gsfc.nasa.gov/data/dataprod/mod09.php accessed on 25 September 2025). The total area of the Tibetan Plateau is accounting for about 1/4 of China's total area. It requires 8 MODIS tiles to achieve complete coverage, as shown in Figure 1. The

corresponding MODIS tiles are h23v05, h24v05, h24v06, h25v05, h25v06, h26v05, h26v06, and h27v06.

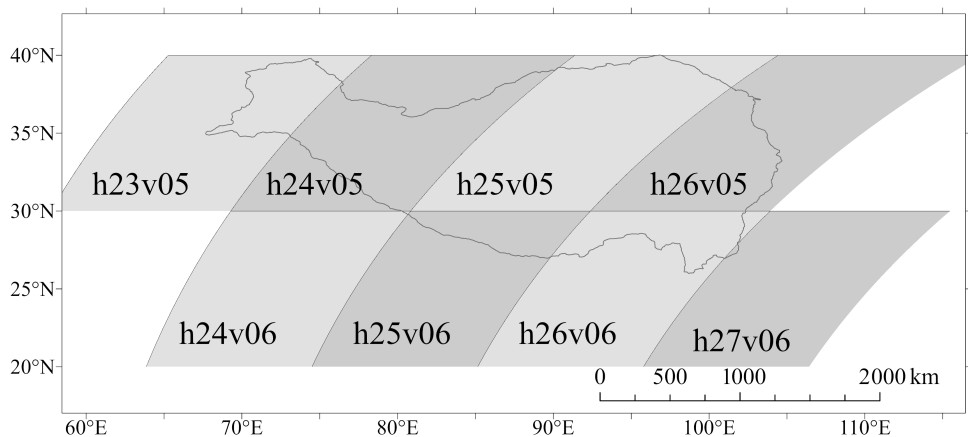

**Figure 1.** Spatial distribution of 8 MODIS tiles covering the Tibetan Plateau.

There are two types of MODIS daily reflectance time series, differing in spatial resolution and spectral band composition: (1) MOD09GQ/MYD09GQ, which includes two bands, the red (R) and NIR band, with a spatial resolution of 250 m and a size of 4800 pixels × 4800 pixels; (2) MOD09GA/MYD09GA, which includes 7 bands from visible (VIS) to SWIR, with a spatial resolution of 500 m and a size of 2400 pixels × 2400 pixels (Table 2). As shown in Figure 2, the spatial resolution of MOD09GQ imagery is higher than that of MOD09GA. So, incorporating MOD09GQ, even though it contains only two bands, can enhance the clarity of the water mapping result.

**Table 2.** Band descriptions of MOD09GQ/MYD09GQ and MOD09GA/MYD09GA.

| Data Name | Spatial Resolution (m) | Band Name | Spectral Range (nm) |
|---|---|---|---|
| MOD09GQ MYD09GQ | 250 | R NIR1 | 620–670 841–876 |
| MOD09GA MYD09GA | 500 | B G R NIR1 NIR2 SWIR1 SWIR2 | 459–479 545–565 620–670 841–876 1230–1250 1628–1652 2105–2155 |

Among them, MOD09GQ and MOD09GA come from MODIS Terra satellite with a 10:30 a.m. equatorial crossing, covering the time range from 24 February 2000 to 31 December 2024, while MYD09GQ and MYD09GA come from the MODIS Aqua satellite with a 1:30 p.m. equatorial crossing time, covering the time range from 4 July 2002 to 31 December 2024. The MODIS reflectance products all use sinusoidal projection. Besides, MOD09GA/MYD09GA also includes a 1-km State QA layer to indicate whether a pixel contains cloud, cloud shadow, snow, aerosol, and so forth.

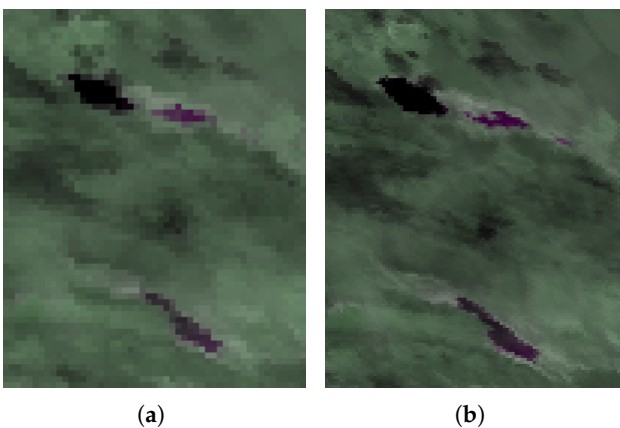

**Figure 2.** The reflectance data of MOD09GA and MOD09GQ (False color image, R: NIR1, G: R, B: NIR1, Year = 2014, DOY = 274, HV = h25v05). (**a**) MOD09GA. (**b**) MOD09GQ.

*2.3. Daily Land Surface Temperature Time Series*

Owing to the high altitude of the Tibetan Plateau, characterized by long winters and low temperatures, the majority of lakes in the region remain frozen for a long duration. LST is important for distinguishing water from low-reflectance snow/ice-cover area. The MODIS LST daily product, MOD11A1 (https://modis.gsfc.nasa.gov/data/dataprod/mod11.php accessed on 25 September 2025) at 1 km spatial resolution, is utilized as an additional information for water and snow/ice classification. The "LST_Day_1km" band, with a spatial resolution of 1000 m and a size of 1200 pixels × 1200 pixels is selected.

*2.4. Topographic Data*

TP has numerous mountains, many of which are covered by snow/ice all year around. The reflectance of mountain shadows is relatively low, which is easily misclassified as water, especially those covered by snow/ice. To eliminate the influence of mountain shadows, the slope data is firstly used to filter out mountainous areas. The slope is typically calculated using Digital Elevation Model (DEM) data. Here, we select the high-quality 30 m Shuttle Radar Topography Mission (SRTM) DEM data (https://srtm.csi.cgiar.org/ accessed on 25 September 2025) to generate slope information.

## 3. Method

The overall flowchart for daily water mapping of the Tibetan Plateau is shown in Figure 3. It mainly contains five steps:

- Preprocessing. The daily cloud, cloud shadow and snow/ice are firstly extract for each pixel based on the MOD09GA/MYD09GA and MOD11A1 time series. In addition, the mountainous areas are also extracted.
- Perpixel water mapping. Perpixel water mapping is performed for each MOD09GA/MYD09GA imagery. The spectral characteristics of water are primarily utilized in this step.
- Superpixel water mapping. Firstly, the segmentation is performed for each MOD09GQ/MYD09GQ scene to get the superpixel result. Then, for each superpixel, a voting based strategy is applied to decide whether it is water or not.
- Fusion. To improve the reliability of daily water mapping, results derived from MOD09GA+MOD09GQ and MYD09GA+MYD09GQ imagery are fused.
- Time-series processing. Cloud, cloud shadow, and missing pixels are interpolated using temporal information, resulting in a final cloud-free daily water mapping product.

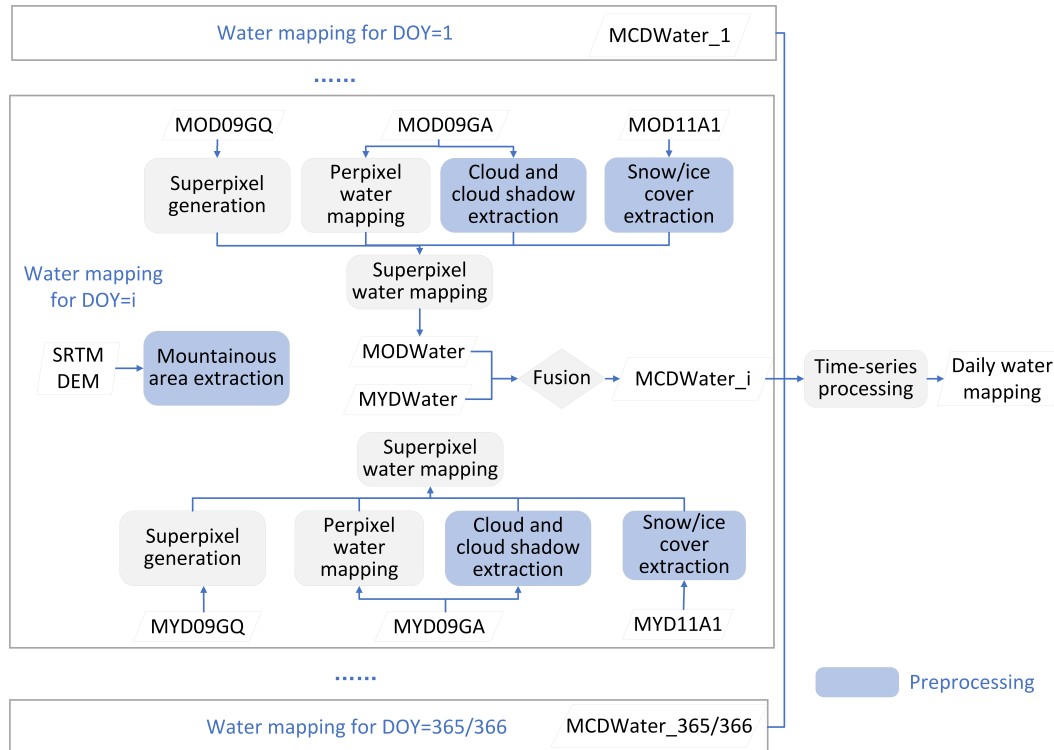

**Figure 3.** The flowchart of daily water mapping based on MOD09QG/MYD09GQ and MOD09GA/MYD09GA time series.

### 3.1. Preprocessing

### 3.1.1. Cloud and Cloud Shadow Extraction

The MODIS State QA layer contain two cloud flags: the cloud state and internal cloud flag. For a given pixel, if either of these two cloud flags is labeled 1, the pixel is identified as a cloud pixel. Based on this criterion, we calculate the annual cloud cover probability for the pixel, as shown in Figure 4. We can see that the two flags over estimated the cloud on water-land boundaries and mountainous areas. Therefore, it is necessary to filter out pixels that have been misclassified as cloud.

The clear-sky method [39] is applied to identify actual cloud pixels. For any pixel located at $(i, j)$ on day-of-year (DOY) $t$ in year $y$, if it is labeled as "cloud" by the MODIS State QA layer, let its reflectance be $\rho$, and the corresponding clear-sky reflectance be $\bar{\rho}$. The pixel is finally confirmed as a cloud pixel if it satisfies the following condition:

$$\left(\rho_{i,j,t,y} - \bar{\rho}_{i,j,t,y}\right)/\bar{\rho}_{i,j,t,y} \geq T_{\mathrm{c}} \tag{1}$$

For any pixel $\rho_{i,j,t,y}$, its clear-sky reflectance $\bar{\rho}_{i,j,t,y}$ is calculated by the median reflectance of all MOD09GA or MYD09GA reflectance within the spatiotemporal window $[i - \Delta i_1 : i + \Delta i_1, j - \Delta j_1 : j + \Delta j_1, t - \Delta t_1 : t + \Delta t_1, y - \Delta y_1 : y + \Delta y_1]$ that are not labeled as "cloud" or "cloud shadow" according to MODIS State QA layer. Here, the red band is selected and parameters are set as $\Delta i_1 = 1$, $\Delta j_1 = 1$, $\Delta t_1 = 7$, $\Delta y_1 = 2$. In addition, since the spatial resolution of the MODIS State QA layer is 1 km, the reflectance in Equation (1) was also downsampled to 1 km for consistency and computational efficiency. Figure 5 presents a illustration of the clear-sky method. A pixel located at the edge of Namtso Lake is selected as an example. The pixel is labeled as "cloudy" by MODIS State QA layer for most of days within a year. However, after applying the clear-sky method, only those points with high reflectance are identified as "cloud".

Next, for simplicity, any pixel adjacent to confirmed cloud pixels, which is also labeled as "cloud shadow" by the MODIS State QA layer, is also labeled as "cloud shadow".

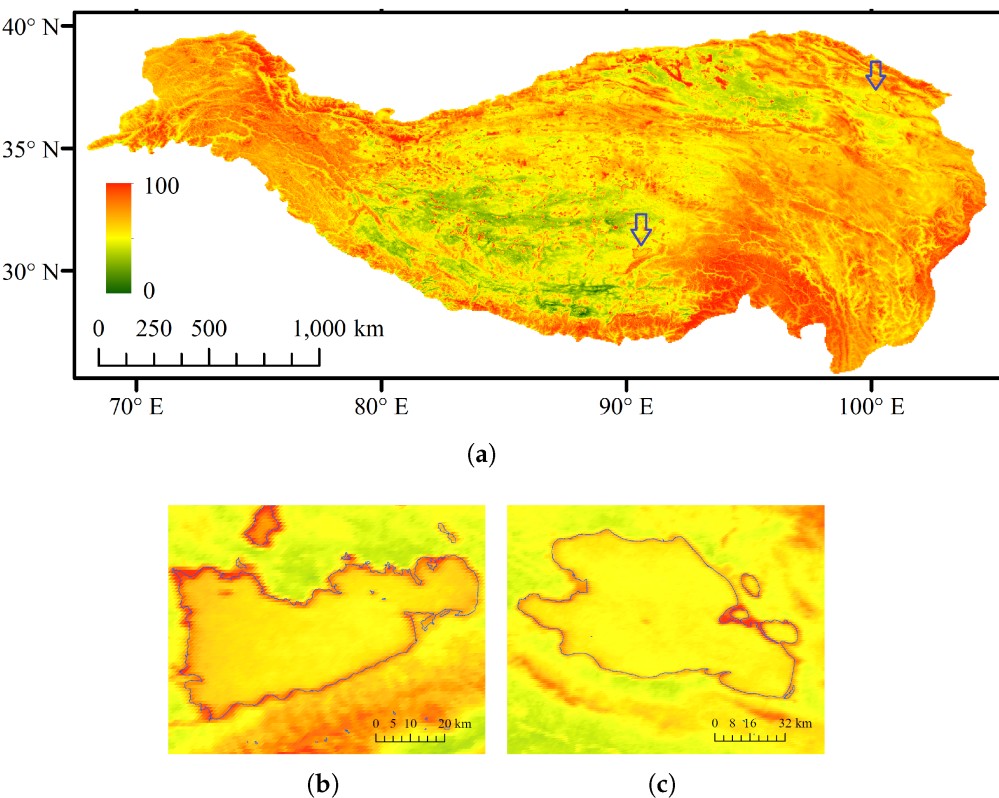

(a)

(b)　　　　　　　　(c)

**Figure 4.** The cloud cover percentage of the Tibetan Plateau in 2015. (**a**) The entire Tibetan Plateau. (**b**) Namtso Lake. (**c**) Qinghai Lake. (Locations of Namucuo Lake and Qinghai Lake are indicated by arrows).

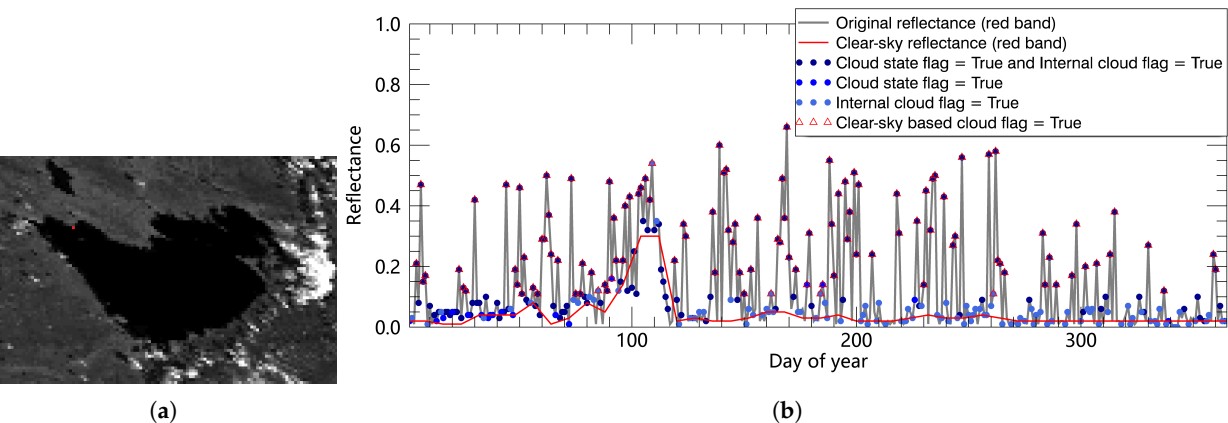

(a)　　　　　　　　　　　　　　　　　(b)

**Figure 5.** Illustration of cloud extraction based on sky-clear method. (**a**) The Namtso Lake (Red band, Year = 2015, DOY = 272). (**b**) The cloud flag derived from the MODIS State QA layer and cloud-sky method.

### 3.1.2. Snow/Ice Cover Extraction

The MODIS QA layer contain two ice/snow flags: the MOD35 snow and internal snow flag. However, those flags greatly underestimate the freezen pixels for lakes. Moreover, they may mistakenly label some pixels in the tropical area as ice/snow [19]. Therefore, in this paper, we abandon these two flags and extract the ice/snow cover according to the change trend of the LST and reflectance values. Due to the existence of cloud, MODIS LST time series is spatially and temporally inconsistent. The Frequency spectrum-Modulated Tensor Completion (FMTC) [11] method is used to generate the cloud free LST, as shown in Figure 6.

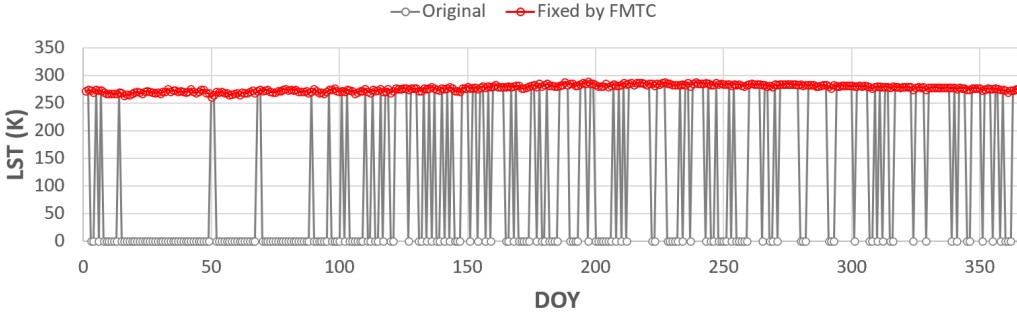

**Figure 6.** The LST time series fixed by FMTC.

Then we calculate the maximum possible extent of freezing throughout a year from the perspective of LST. For pixel $(i, j, t, y)$, we firstly compute the percentage of LST within a temporal window below $LST_0$, that is

$$p_{i,j,t,y}^{\text{LST}} = \frac{N_{i,j,t,y}^{\text{LST}}}{N} \tag{2}$$

where $N_{i,j,t,y}^{\text{LST}}$ is the number of pixels that satisfy $LST_{i,j,t-\Delta t_2:t+\Delta t_2,y} \leq LST_0$, and $N = 2 \times \Delta t_2 + 1$. As DOY increasing from 1 to 365/366, if the pixel satisfies

$$p_{i,j,t,y}^{\text{LST}} \geq T_{\text{LST}} \tag{3}$$

then it is labeled as "snow/ice (possible)". Then for the pixel $(i, j, t, y)$ labeled as "snow/ice (possible)" and not as "cloud", if it satisfies

$$\rho_{i,j,t,y} - \tilde{\rho}_{i,j,y} \geq T_{\text{ICE}} \tag{4}$$

then it is identified as "snow/ice", where $\tilde{\rho}_{i,j,y}$ is calculated by the average reflectance of the pixel $(i, j)$ that is not not labeled as "cloud" or "cloud shadow" of the summer season for the $y$th year. The reflectance used in Equation (4) is the same 1 km reflectance used in Equation (1) According to our experience and statistical analysis, we set $LST_0 = 273.15$ K, $T_{\text{LST}} = 0.7$, and $T_{\text{ICE}} = 0.15$.

### 3.1.3. Mountainous Area Extraction

Based on the report on the geographical characteristics of China's wetland for 2000, wetlands with slope less than 3° and 8° occupy 93.85% and 99.17% of the total wetland areas, respectively [40]. In this report, wetland includes river, lake, reservoir/pond, and urban/entertainment water. That is to say, water are usually distributed on flat terrain. Therefore, in our study, pixels with slopes greater than 8° are labeled as "Mountainous area" [40].

### 3.1.4. Perpixel Water Mapping Based on 500 m MODIS Reflectance Time Series

Generally, water has a higher reflectance in VIS-NIR range than that in SWIR range [18]. Many water indices have been designed based on this spectral characteristic. When the threshold-based water indices were applied to images of different regions acquired at different times, it is challenging to determine an appropriate threshold to accurately discriminate water and non-water pixels in complex background [41]. Therefore, the band comparing water index (BCWI) [19] which does not require threshold, is used here for water mapping,

$$BCWI_{i,j,t,y} = \begin{cases} 0 \text{ (non-water)}, & \text{if } \rho_{i,j,t,y}^{\text{maxVIS}} < \rho_{i,j,t}^{\text{maxSWIR}} \\ 1 \text{ (water)}, & \text{if } \rho_{i,j,t,y}^{\text{maxVIS}} \geq \rho_{i,j,t}^{\text{maxSWIR}} \end{cases} \tag{5}$$

where $\rho_{i,j,t,y}^{\text{maxSWIR}} = \max\left\{\rho_{i,j,t,y}^{\text{SWIR1}}, \rho_{i,j,t,y}^{\text{SWIR2}}\right\}$, meaning the maximal reflectance in SWIR bands. However, due to the atmospheric correction error, the reflectance of very dark water may not exhibit this property. Therefore, if a pixel satisfies

$$\rho_{i,j,t,y}^{\text{maxVIS}} \leq T_{\text{LW}} \text{ and } \rho_{i,j,t,y}^{\text{maxSWIR}} \leq T_{\text{LW}} \tag{6}$$

it is also labeled as "water". The threshold $T_{\text{LW}}$ is set to 0.05 [19].

### *3.2. Superpixel Water Mapping*

Superpixel-based mapping can overcome the problem of salt-and-pepper effect and improve the precision of delineated boundaries. Therefore, we use the 2-band MODIS 250 m reflectance data to obtain superpixels, and then a voting-based method is used for each superpixel to determine the land cover type.

### 3.2.1. Superpixel Generation Based on 250 m MODIS Reflectance Time Series

The unsupervised classfication method, K-means-based Connection Center Evolution (KCCE) is used to obtain superpixels [21]. CCE is proposed to achieve mutli-scale classification [42]. In CCE, a point is considered a cluster center if its self-connectivity is stronger than its connectivity to any other point. Remaining data points are then assigned to these centers according to relative connectivity, enabling multi-scale clustering without prior knowledge of the number of clusters. Yet, directly applying CCE to large-scale images is computationally prohibitive because constructing a full similarity matrix requires excessive memory.. To solve this problem, KCCE replaces the original large-size data points with a few representative sets, which is achieved by applying K-means firstly, then uses the center point of each set to construct the similarity matrix. KCCE significantly reduces computational and memory costs, making it feasible for large-scale images.

Specifically, we first apply K-means to divide each tile of MOD09GQ into 101 clusters and compute the mean spectrum of each cluster. These representative mean vectors are then used as inputs to the CCE classifier for further clustering. Through its iterative evolution process, CCE produces multi-scale classification results, ranging from $N$ clusters (where $N$ is the number of pixels) down to a single cluster. In practice, we typically select the result with about 10 clusters as the final output. The resulting partitions obtained by KCCE are then sequentially labeled as individual regions, where each region is regarded as one superpixel, as shown in Figure 7.

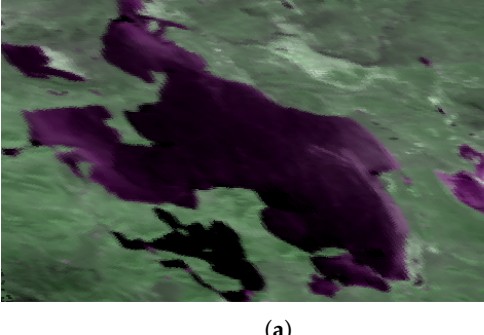

(**a**)　　　　　　　　　　　　　　(**b**)

**Figure 7.** *Cont.*

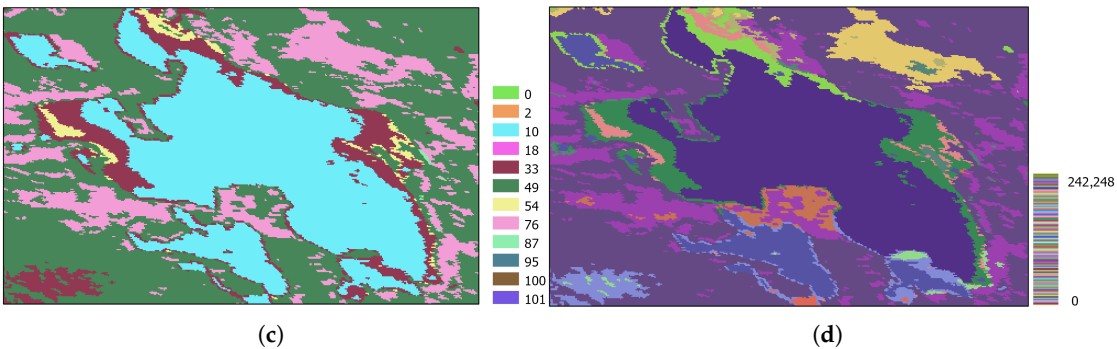

**Figure 7.** The process of generating superpixels using KCCE. (**a**) MOD09GQ (false color composite, R: NIR1, G: Red, B: NIR1, 250 m, acquisition time: Year = 2010, DOY = 083); (**b**) K-means clustering result with 101 classes; (**c**) KCCE clustering result with 12 classes; (**d**) Superpixel labeling.

3.2.2. Voting-Based Strategy for Superpixel Water Mapping

For each superpixel, we firstly calculate the following proportions,

- Water proportion: the fraction of pixels within the superpixel that are classified as "water".
- Snow/Ice proportion: the fraction of pixels labeled as "Snow/Ice".
- Cloud proportion: the fraction of pixels labeled as "cloud".
- Cloud shadow proportion: the fraction of pixels labeled as "cloud shadow".
- Mountainous area proportion: the fraction of pixels labeled as "mountainous area".

After obtaining these proportions, the majority voting strategy is applied for each superpixel. That is, the class of each superpixel is assigned according to the class with the highest proportion is assigned.

*3.3. Fusion*

Since both MODIS Terra and Aqua data are utilized, each pixel has two corresponding mapping results. To integrate them and obtain a more stable and reliable outcome, we applied the following fusion strategy:

- When the two datasets give the same classification result, that value is directly adopted.
- When one dataset indicates cloud, cloud shadow, or mountainous area while the other provides a valid classification, we adopt the valid classification result.
- When both datasets provide valid but different classifications, then the distance between sensor and earth is applied, which is recorded as "Range" layer in MOD09GA/MYD09GA dataset. MODIS uses a whiskbroom scanning imaging mode, so imagery acquired at the nadir has the highest spatial resolution and clarity with the minimal range value. In contrast, pixels at the scan edges are captured at higher viewing angles, leading to a larger range value and lower image sharpness, as shown in Figure 8. Therefore, the classification result with the smaller range value is selected as more reliable.

This rule-based fusion ensures consistency between Terra and Aqua observations and improves the robustness of the final water mapping product.

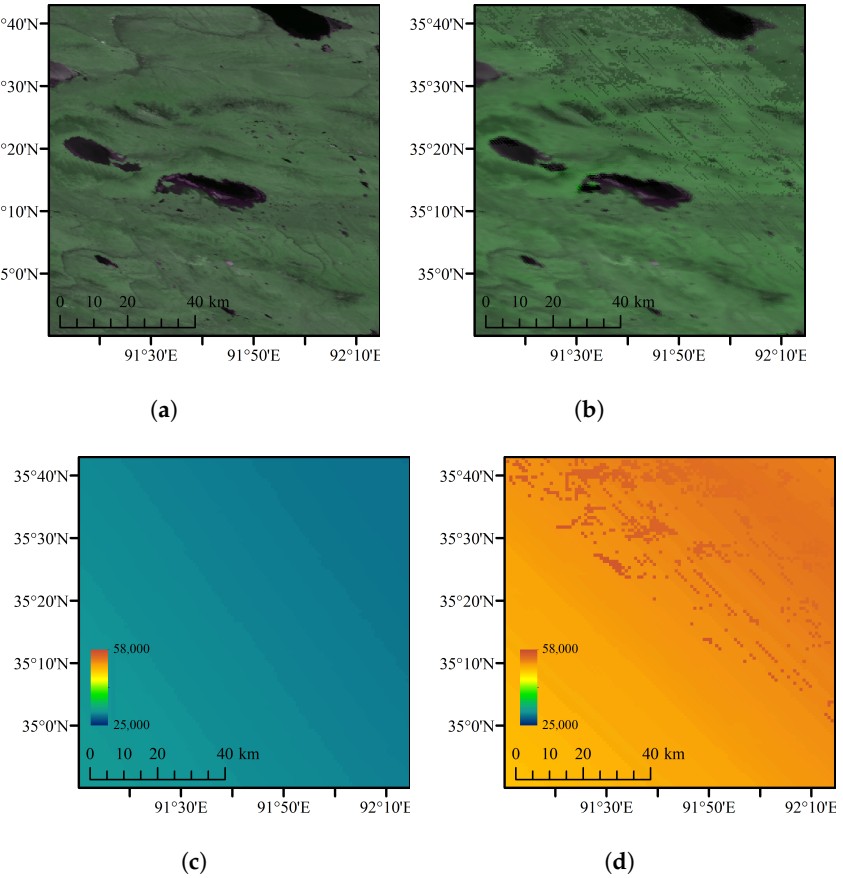

**Figure 8.** The MOD09GA and MYD09GA imagery at the same day (Year = 2024, DOY = 138) with different ranges. (**a**) MOD09GA, (**b**) MYD09GA, (**c**) Range distribution corresponding to MOD09GA imagery, and (**d**) Range distribution corresponding to MYD09GA imagery. Clearly, the range of the MOD09GA image are much smaller than that of the MYD09GA image, so the MOD09GA image has a higher clarity.

### 3.4. Time-Series Processing

Despite the fusion of MODIS Terra and Aqua data, the results still contain cloud, cloud shadow, and even missing pixels. Thus, time series processing is required to obtain the cloud-free water mapping result. In this study, we adopt the temporal nearest-neighbor interpolation approach. That is, if a pixel is is labeled as "cloud" or "cloud shadow", then the nearest valid observation within $\Delta T$ days is selected for interpolation. Here, $\Delta T$ is set according to the refletance variety of a pixel, which is named the dynamic index,

$$D_{i,j,t} = \frac{\sigma_{i,j,t}}{\mu_{i,j,t}} \tag{7}$$

where $\sigma_{i,j}$ is calculated as the standard deviation of all MOD09GA or MYD09GA reflectance within the temporal window $[t - \Delta t_3 : t + \Delta t_3, y = 2000 : 2024]$ that are not labeled as "cloud" or "cloud shadow". Correspondingly, $\mu_{i,j,t}$ represents the mean reflectance computed over the same subset. In this study, we set $\Delta t_3 = 15$. Based on the value of the dynamic index $D_{i,j}$, different thresholds $\Delta T$ are set as follows:

- When $D \in [0, 0.1)$, the pixel is considered as low dynamic, and we set $\Delta T = 180$. If no valid data is found within $\Delta T$ days, annual-aggregation results are used for gap-filling.
- When $D \in [0.1, 0.2)$, the pixel is considered moderately dynamic, and we set $\Delta T = 30$. If no valid data is found within $\Delta T$ days, seasonal-aggregation results are used for gap-filling.

- When $D \in [0.2, \infty)$, the pixel is considered highly dynamic, and we set $\Delta T = 15$. If no valid data is found within $\Delta T$ days, monthly-aggregation results are used for gap-filling.

## 4. Results

### 4.1. Spatial-Temporal Distribution of Water Bodies in Tibetan Plateau

Figure 9 shows the number of water cover days across the plateau for the year 2002, 2012, and 2022, along with zoomed-in views of Chaerhan Salt Lake and Selin Co. We can see that our product can effectively capture lake expansion and spatial dynamics. Figure 10 presents the annual, monthly, and daily water area curves for Selin Co, showing that the our product can reflect both interannual and intrannual variations in the extent of lakes on the Tibetan Plateau.

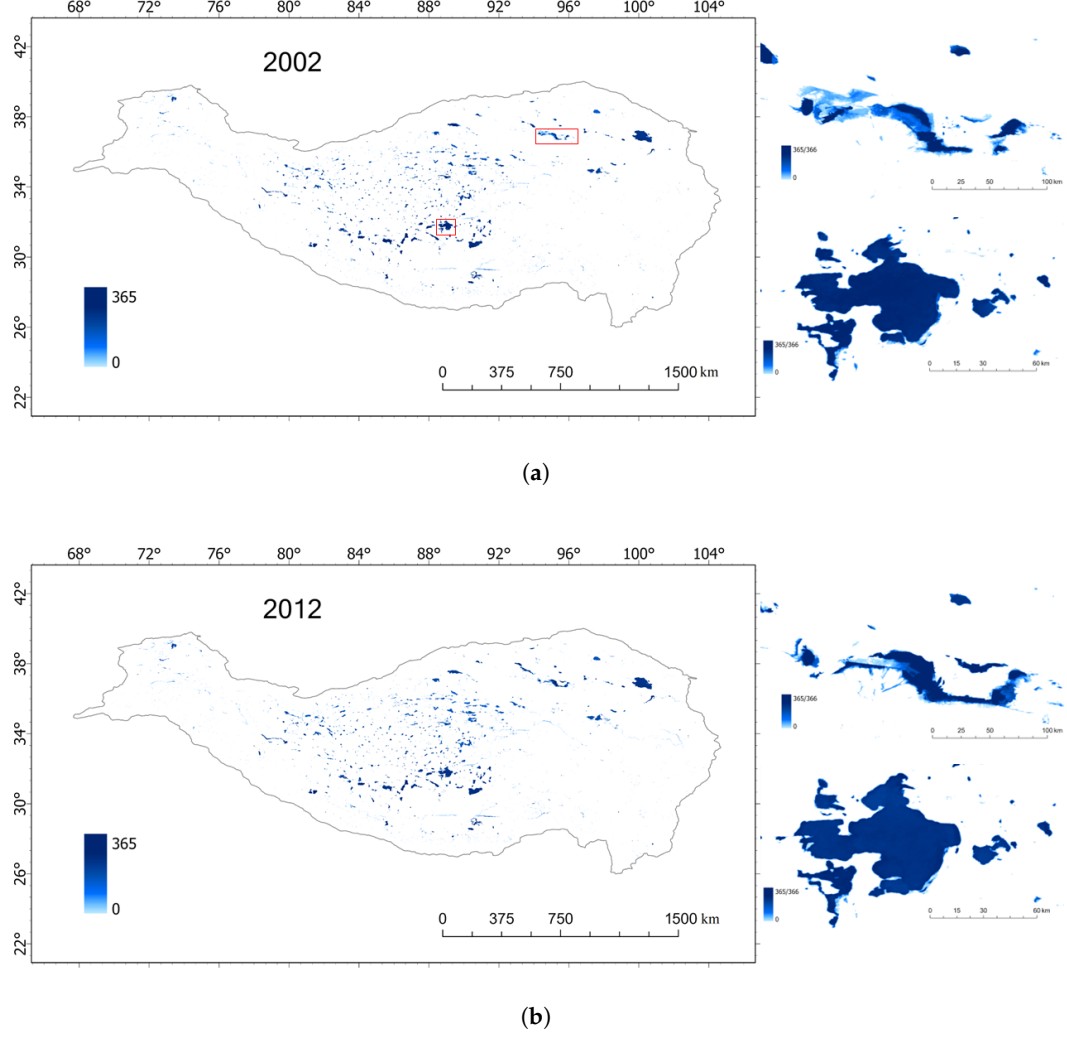

(**a**)

(**b**)

**Figure 9.** *Cont.*

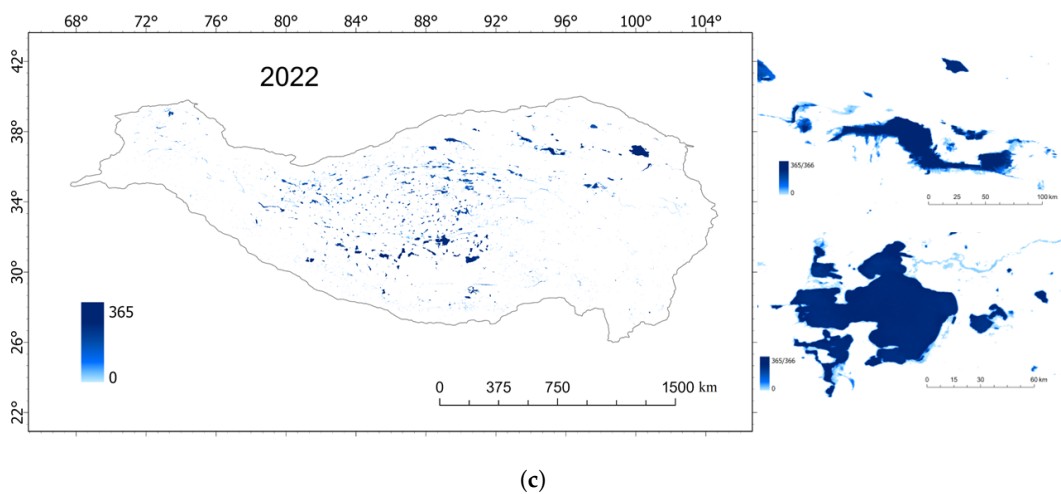

(**c**)

**Figure 9.** Water cover days of the Tibetan Plateau for (**a**) 2002, (**b**) 2012, and (**c**) 2022.

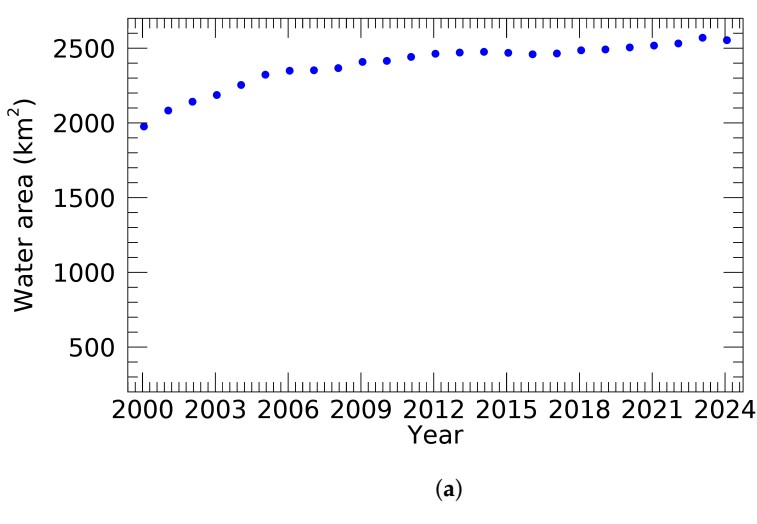

(**a**)

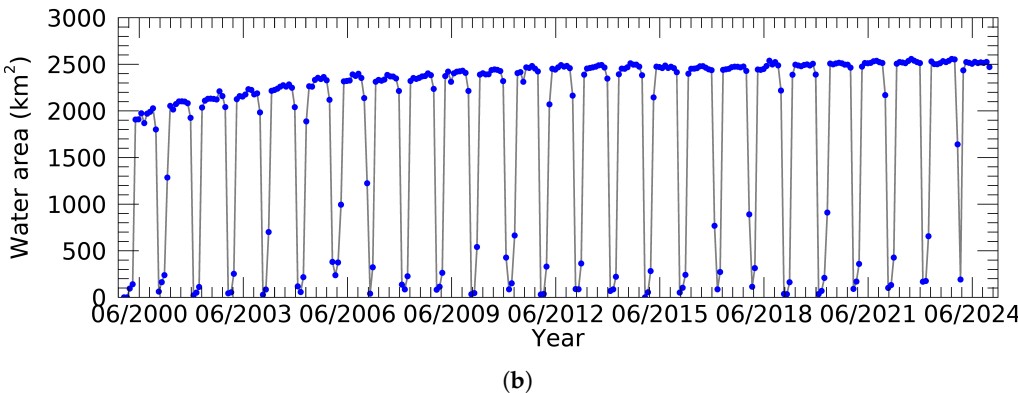

(**b**)

**Figure 10.** *Cont*.

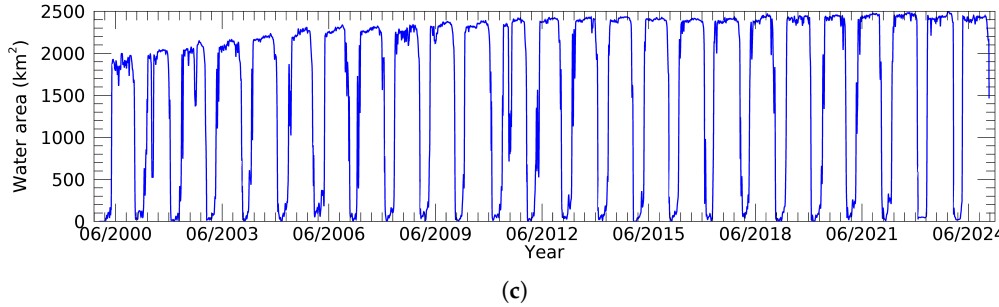

(**c**)

**Figure 10.** The yearly (**a**), monthly (**b**), and daily (**c**) water area curves of Selin Co.

*4.2. Validation*

In this study, we employ two approaches to validate the accuracy of our product: (1) direct validation, using water mapping results from quasi-synchronous high-resolution satellite imagery, such as the Landsat image; (2) cross-validation, using results from other high-resolution mapping products, such as the 30 m global water mapping produced by the Joint Research Centre (JRC-Water) [35]. For the direct validation, a confusion matrix is used to calculate the recall, precision, and overall accuracy [19] for land, water, and snow/ice (within water body areas only). For cross-validation, the root mean square error (RMSE), mean relative error (MRE), and $R^2$ are calculated based on the water areas derived from our product and the other high-resolution product.

### 4.2.1. Direct-Validation with High-Resolution Water Mapping Result

Between 2000 and 2024, 38 randomly selected Landsat scenes (Figure 11) were manually interpreted to obtain the ground truth of land, water, and snow/ice (limited to water areas). A confusion matrix was then calculated, as shown in Table 3. It can be seen that our product has a high accuracy in water mapping, with both recall and precision exceeding 95%. The accuracy for snow/ice on water areas is also high, reaching over 98%, but the recall is relatively low, at around 81%. This indicates that a considerable number of frozen pixels were misclassified as water (Table 3). A possible reason is that the ice information in this product is derived from a land surface temperature product with a spatial resolution of 1000 m, which makes it easy to miss small patches of lake ice. Consequently, the recall for snow/ice on water bodies is relatively low.

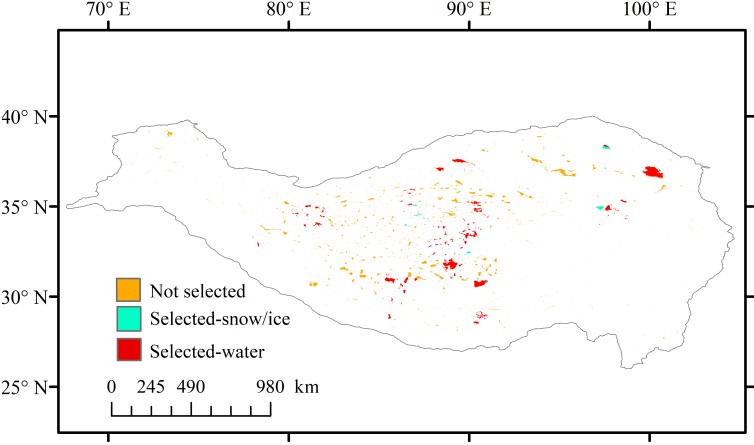

**Figure 11.** Locations of lakes selected for the direct validation.

**Table 3.** Confusion matrix based on water mapping results from Landsat images.

| | | **Ground Truth** | | | |
|---|---|---|---|---|---|
| | | Land | Water | Snow/Ice | **Precision (%)** |
| **Classified** | Land | 883,705 | 17,036 | 5771 | 97.4841 |
| | Water | 15,925 | 535,928 | 7383 | 95.8322 |
| | Snow/Ice | 794 | 357 | 57,583 | 98.0403 |
| | **Recall (%)** | 98.1432 | 96.8566 | 81.4044 | 96.8995 |

### 4.2.2. Cross-Validation with Other Water Mapping Product

Here, we use the annual water body product from JRC Water [35] for comparison. Across the entire Tibetan Plateau, we randomly selected 44 lakes, as shown in Figure 12, and calculated the lake areas derived from both products. The results are presented in Table 4 and Figure 4.

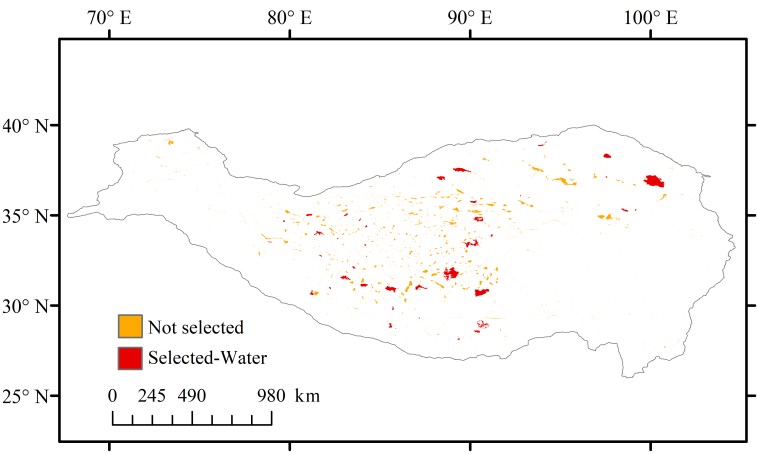

**Figure 12.** Locations of lakes selected for the cross-validation.

**Table 4.** Comparison results based on water mapping results from JRC Water products.

| **Size** | **RMSE (km$^2$)** | **MRE (%)** | **R$^2$** |
|---|---|---|---|
| $\geq$2000 km$^2$ | 173.4160 | 3.8687 | 0.996342 |
| 1000$\sim$2000 km$^2$ | 93.6652 | 6.9041 | 0.951405 |
| 500$\sim$1000 km$^2$ | 65.2778 | 6.9961 | 0.945332 |
| $\leq$500 km$^2$ | 13.8417 | 10.2867 | 0.997600 |
| All | 59.5831 | 9.0526 | 0.999258 |

We can see that as the lake area decreases, the RMSE gradually decreases while the MRE increases, indicating that the larger the lake is, the higher the accuracy of our product is. This is mainly because our product has a much coarser spatial resolution (250 m) than JRC Water. As a result, mapping small-sized water accurately is more challenging for our product. Overall, as shown in Figure 13, the lake area estimates from our product and the JRC Water annual product exhibit a high level of consistency ($MRE \leq 10\%$, $R^2 \geq 0.99$).

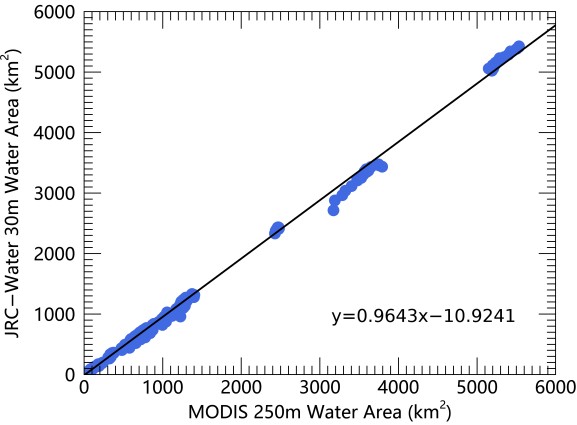

**Figure 13.** Comparison of lake areas between the JRC Water product and our product.

### 4.3. Change Trend for Lakes the Tibetan Plateau

On the Tibetan Plateau, the predominant type of water is the lake. Based on [43], we are able to extract 1293 lakes (with area $\geq 5$ km$^2$) from our 250 m product. In the following analysis, we investigate the spatiotemporal variation trends of these lakes from 2000 to 2024. The spatial distribution of the annual change rate of lake areas on the Tibetan Plateau are shown in Figure 14. It can be observed that (1) the number of lakes with an increasing rate significantly exceeds those with a decreasing rate and that (2) the fastest change rate of increase is $+43.18$ km$^2$/y, whose absolute value is much larger than that of the fastest rate of decrease ($-5.91$ km$^2$/y). Specifically, as tabulated in Table 5, over the 25-year period, 87.70% (1134 out of 1293) of lakes experienced an increase in area, while only 12.30% (159 out of 1293) showed a decrease. The number of lakes with an annual increase rate $\geq 0.5$ km$^2$/y is 187, which is far greater than the number with an annual decrease rate $\leq -0.5$ km$^2$/y (12 in total). The results indicate that over the past 25 years, lakes on the Tibetan Plateau have generally exhibited a significant expansion trend. Furthermore, the rate of increase for some lakes far exceeds the rate of decrease, highlighting a pronounced overall growth.

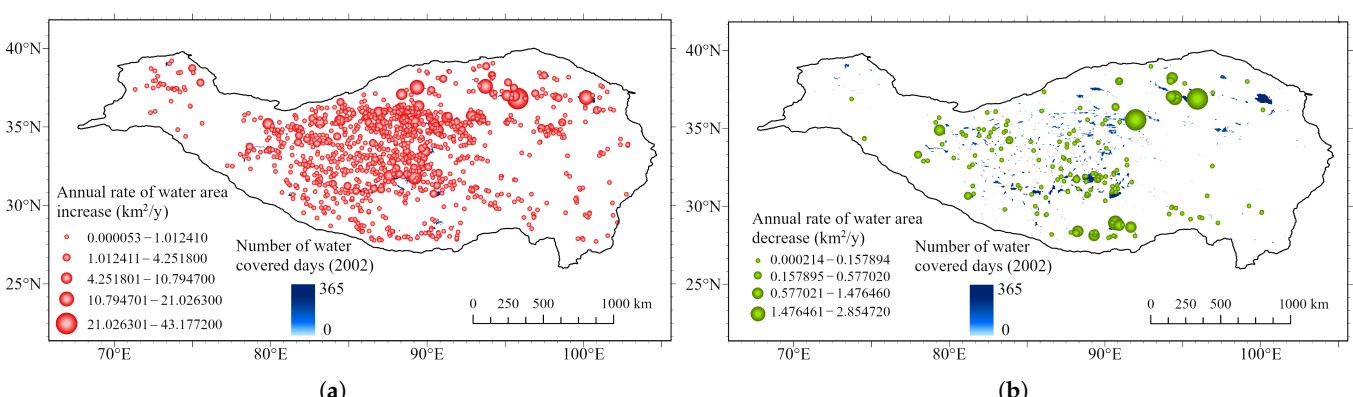

**Figure 14.** Annual change rates of lakes on the Tibetan Plateau from 2000 to 2024. (**a**) Increase rate. (**b**) Decrease rate.

**Table 5.** The distribution of annual water area change rates on the Tibetan Plateau from 2000 to 2024.

| Changing Rate (km$^2$/y) | $\leq -0.5$ | $(-0.5, 0)$ | $[0, 0.5)$ | $\geq 0.5$ |
|---|---|---|---|---|
| **Number of Lakes** | 12 | 147 | 947 | 187 |

### 4.4. Change Trend for the 10 Largest Lakes

From 2000 to 2024, the area of lakes on the Tibetan Plateau underwent significant changes, resulting in a shifting ranking of the 10 largest lakes, as shown in Table 6. While Qinghai Lake consistently remained the largest lake, the rankings from 2nd to 10th changed over time. Nam Co and Selin Co alternated between the second and third positions, with Selin Co surpassing Nam Co in 2001 to become the second-largest lake on the plateau (Figure 15b). As for the lakes ranked from third to tenth, the Qarhan Salt Lake rose steadily from outside the top 10 in 2000 and maintained third position after 2009. In contrast, Yamzho Yumco experienced a continuous decline after 2004, falling out of the top 10 after 2006 (Figure 15c).

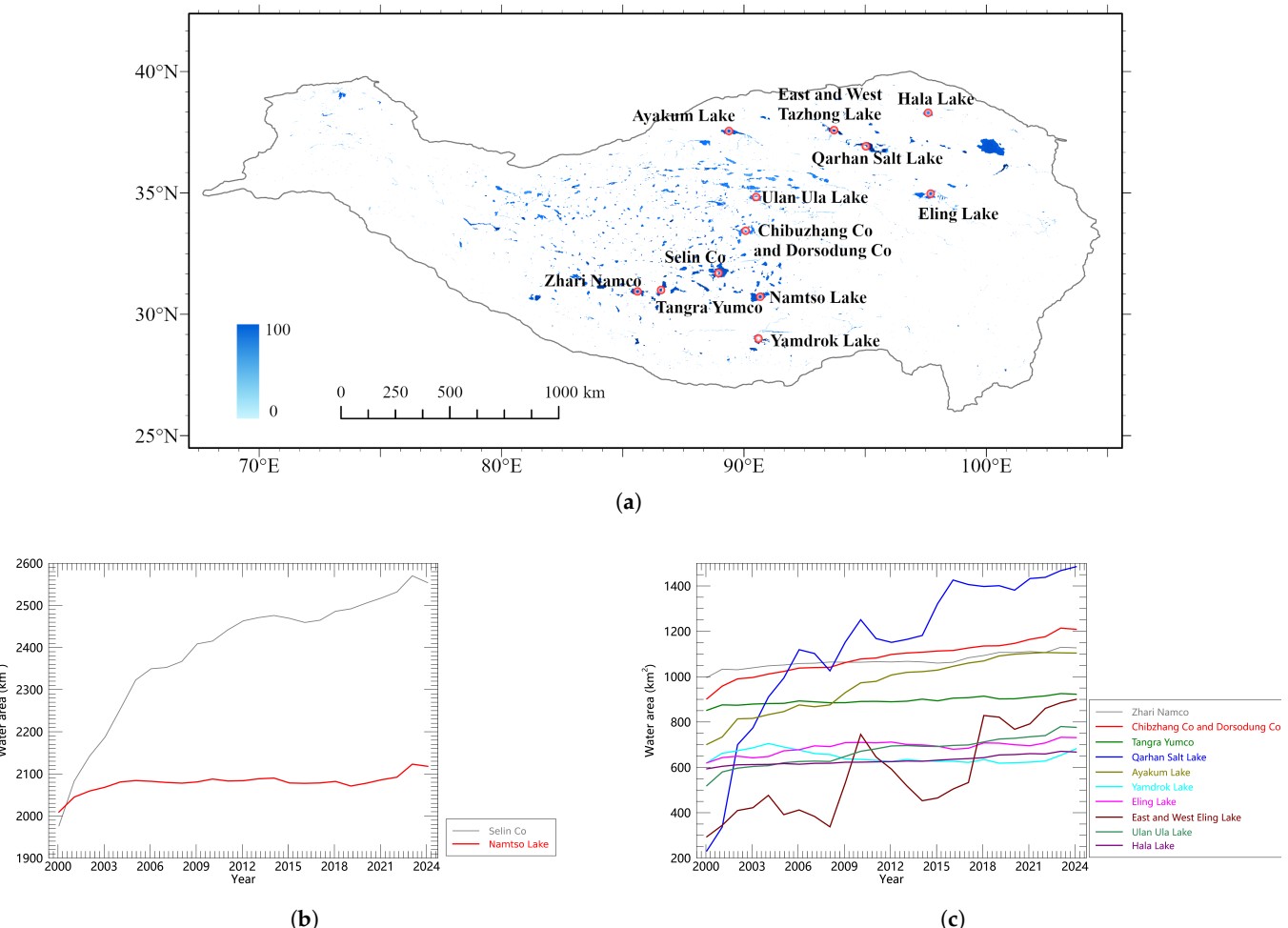

(**a**)

(**b**)        (**c**)

**Figure 15.** Water area curves from 2000 to 2024 for 12 lakes that appeared in the annual top-10 lists. (**b**) Namtso Lake and Selin Co. (**c**) Zhari Namco, Chibuzhang Co and Dorsodong Co, Tangra Yumco, Chaerhan Salt Lake, Ayak Kum Lake, Yamzhog Yumco, Eling Lake, East and West Taijinar Lake, Ulan Ula Lake, and Hala Lake.

**Table 6.** Ten largest lakes on the Tibetan Plateau in 2000, 2005, 2010, 2015, 2020, and 2024.

| No. | 2000 | 2005 | 2010 | 2015 | 2020 | 2024 |
|-----|------|------|------|------|------|------|
| 1 | Qinghai Lake | Qinghai Lake | Qinghai Lake | Qinghai Lake | Qinghai Lake | Qinghai Lake |
| 2 | Namtso Lake | Selin Co | Selin Co | Selin Co | Selin Co | Selin Co |
| 3 | Selin Co | Namtso Lake | Namtso Lake | Namtso Lake | Namtso Lake | Namtso Lake |
| 4 | Zhari Namco | Zhari Namco | Qarhan Salt Lake | Qarhan Salt Lake | Qarhan Salt Lake | Qarhan Salt Lake |

**Table 6.** *Cont.*

| No. | 2000 | 2005 | 2010 | 2015 | 2020 | 2024 |
|---|---|---|---|---|---|---|
| 5 | Chibuzhang Co and Dorsodung Co | Chibuzhang Co and Dorsodung Co | Chibuzhang Co and Dorsodung Co | Chibuzhang Co and Dorsodung Co | Chibuzhang Co and Dorsodung Co | Chibuzhang Co and Dorsodung Co |
| 6 | Tangra Yumco | Qarhan Salt Lake | Zhari Namco | Zhari Namco | Zhari Namco | Zhari Namco |
| 7 | Ayakum Lake | Tangra Yumco | Ayakum Lake | Ayakum Lake | Ayakum Lake | Ayakum Lake |
| 8 | Yamdrok Lake | Ayakum Lake | Tangra Yumco | Tangra Yumco | Tangra Yumco | Tangra Yumco |
| 9 | Eling Lake | Yamdrok Lake | East and West Eling Lake | East and West Tazhong Lake | East and West Tazhong Lake | East and West Tazhong Lake |
| 10 | Hala Lake | Eling Lake | Eling Lake | Ulan Ula Lake | Ulan Ula Lake | Ulan Ula Lake |

*4.5. Seasonal Variation Characteristics of Water Body Area*

Figure 16 shows the frequency distribution of the months in which the minimum and maximum areas of lakes occur on the Tibetan Plateau. It can be seen that the minimum area typically occurs in January, while the maximum area occurs in October. The minimum area in January is primarily due to the freezing of most lakes during the winter season. In contrast, the maximum area in October results from the accumulation of water starting in August, driven by increased precipitation during the rainy season and snow/ice melt caused by rising temperatures. By the end of the rainy season, in October, the lake area reaches its annual peak.

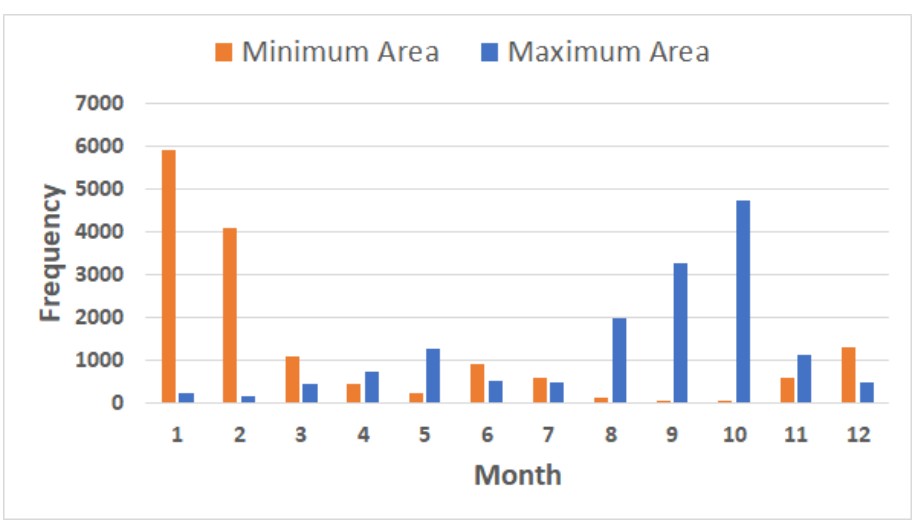

**Figure 16.** Statistical analysis of monthly area of lakes on the Tibetan Plateau.

## 5. Discussion

*5.1. Influence of Spatial Resolution on the Classification of Snow/Ice and Water*

As shown in Table 3, the classification accuracy for snow and ice is relatively lower. The main reason for this is that the datasets used for snow/ice classification are primarily MOD11A1 and the resampled red band from MOD09GA, both of which have a spatial resolution of 1 km. In contrast, the datasets used for water classification provide higher spatial resolution (250 m for MOD09GQ/MYD09GQ and 500 m for MOD09GA/MYD09GA). Consequently, the snow/ice classification is less accurate, particularly for small ice patches and along ice–water boundaries, as illustrated in Figure 17.

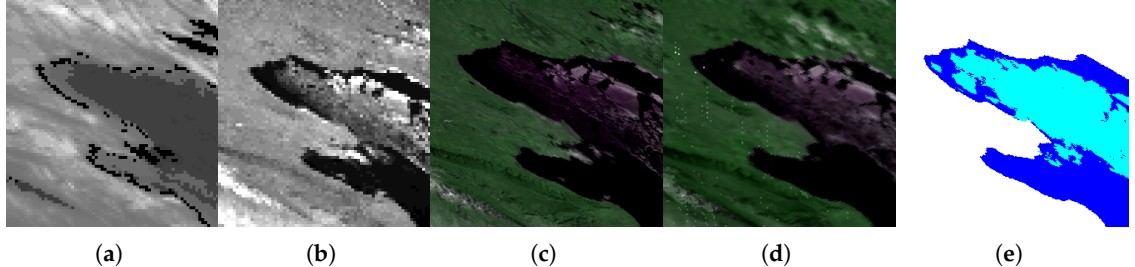

**Figure 17.** Effect of spatial resolution on the classification of snow/ice. (**a**) MOD11A1 LST (1 km); (**b**) MO09GA red band (resampled to 1 km); (**c**) MOD09GQ (false color, R: NIR1, G: R, B: NIR1, 250 m); (**d**) MYD09GQ (false color, R: NIR1, G: R, B: NIR1, 250 m); (**e**) classification result (water—blue; ice—cyan; 250 m). Acquisition time: year = 2010, DOY = 083.

Similarly, although we employed the MODIS product with the highest spatial resolution available (MOD09GQ series), its resolution is limited to 250 m. As a result, the delineation of lake boundaries is not particularly precise. Furthermore, as reflected in Table 4, the extraction accuracy for small water bodies is relatively low, which is also illustrated in Figure 18. Therefore, if higher accuracy in boundary delineation or small water body mapping is required, it is recommended to use higher-resolution remote sensing data, such as Landsat, Sentinel, or the Gaofen satellite series [44]. In addition, integrating these higher-resolution datasets with MODIS time-series data could be a promising approach, as it would leverage both the temporal richness of MODIS and the spatial detail of higher-resolution sensors, thereby improving the robustness of water body monitoring and mapping.

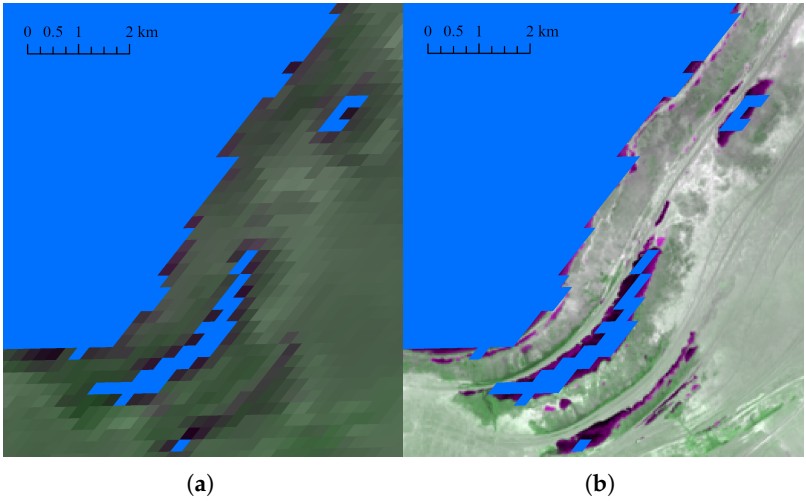

**Figure 18.** Effect of spatial resolution on the classification of water. (**a**) MYD09GQ (false color, R: NIR1, G: R, B: NIR1, 250 m); (**b**) Landsat 8 (false color, R: NIR, G: R, B: NIR). Pixels classified as "water" in our product are colored in blue. Acquisition time: year = 2020, DOY = 296.

### 5.2. Driving Factors Behind the Observed Lake Area Changes

As shown in Figure 14 and Table 5, approximately 87% of the lakes experienced an increase in area. The main driving factors are enhanced precipitation, increased glacier meltwater, and reduced evaporation. Among these, changes in glacier meltwater and evaporation are primarily influenced by temperature variations. In the northwestern part of the Tibetan Plateau, the increase in glacier meltwater has been the dominant factor contributing to lake expansion and water volume rise [45]. Although reduced potential evaporation and increased glacier meltwater may jointly facilitate lake growth, statistical

analysis indicates that the primary factor behind the expansion of most lakes is the increase in regional precipitation [3].

In addition to climatic drivers, human activities have also played a role in accelerating lake area expansion. A prominent example is the Chaaerhan Salt Lake, which originally consisted of natural brine pools located in the lowest and most central part of the Qaidam Basin. Owing to its abundant mineral resources, an industrial salt field was established at the lake's center. With the growing demand for potash fertilizers, the salt field has expanded rapidly, far exceeding the size of the remaining natural lakes shown in Figure 15. This case highlights the significant impact of industrial development on lake evolution, in contrast to purely climate-driven processes.

By contrast, a small proportion of lakes exhibited an overall decrease in area during the 25-year study period. Typical examples include Yamdrok Lake and Zhuonai Lake. The shrinkage of Yamdrok Lake since 2000 is mainly due to human activity through the construction and operation of a hydropower station, which started to play a strong role after 1998 [46]. For Zhuonai Lake, a dramatic reduction occurred on 14 September 2011, when continuous heavy rainfall caused the lake to overflow and breach its banks, leading to a sudden outburst flood and rapid decline in lake area.

Overall, these findings demonstrate that climate variability—particularly precipitation and glacier meltwater—has been the dominant driver of lake dynamics on the Tibetan Plateau, while localized human activities can also exert significant and sometimes decisive impacts on individual lakes. In the future, incorporating studies on water level or volume changes [47] would be more beneficial for the quantitative assessment of the driving factors.

## 6. Conclusions

This study developed and validated a comprehensive daily surface-water mapping dataset for the Tibetan Plateau covering the period of 2000–2024, explicitly accounting for both water bodies and ice cover dynamics. Leveraging MODIS daily reflectance and LST products, combined with spectral analysis, superpixel segmentation, and temporal interpolation, we achieved an overall accuracy of 96.89% and a mean relative error of less than 9.1%. This robust dataset enables the long-term, cloud-free, daily monitoring of lakes across the Plateau, supporting investigations of both long-term interannual changes and short-term seasonal fluctuations.

The scope of our analysis focused on 1293 lakes larger than 5 km$^2$, revealing that nearly 88% of lakes expanded over the past 25 years, whereas a smaller fraction exhibited shrinkage, often linked to human activities or abrupt hydrological events. Seasonal variations showed consistent patterns, with lake areas peaking in October and reaching minima in January.

Future research should aim to integrate multi-sensor data (e.g., Sentinel-1/2, Landsat) to enhance both spatial and temporal accuracy, and to incorporate altimetry- or gravimetry-based datasets for assessing lake water levels and storage changes. Additionally, coupling the dataset with climate and socio-economic data could provide deeper insights into the interactions among climate variability, human activities, and water resource sustainability on the Tibetan Plateau.

**Author Contributions:** Conceptualization and writing, Q.F.; investigation and validation, K.Y.; methodology, L.J. All authors have read and agreed to the published version of the manuscript.

**Funding:** This research was funded by the National Key Research and Development Program of China (grant number: 31400).

**Data Availability Statement:** The water mapping result is available at https://doi.org/10.11888/Terre.tpdc.

**Conflicts of Interest:** The authors declare no conflicts of interest.

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
