# Peer review of "Daily Water Mapping and Spatiotemporal Dynamics Analysis over the Tibetan Plateau"

_hydrology, doi:10.3390/hydrology12100257_

Round 1
Reviewer 1 Report
Comments and Suggestions for Authors
The paper presents a novel daily surface water mapping dataset for the entire Tibetan Plateau covering the period 2000–2024, leveraging both Terra and Aqua MODIS reflectance (250 m and 500 m) and Land Surface Temperature products to capture liquid water and ice phenology. Its main contributions include (1) integrating multiple MODIS bands and sensors to mitigate cloud and shadow effects, (2) combining pixel- and superpixel-level classification with fusion and temporal interpolation to deliver a gap-free daily water map, and (3) providing a comprehensive 25-year spatiotemporal analysis of lake dynamics—quantifying interannual trends, seasonal patterns, and change rates for over 1,200 lakes. The strengths of this work lie in its high temporal resolution, rigorous validation against Landsat and JRC-Water products (achieving >96% accuracy), and its detailed methodological framework that effectively balances spatial detail and temporal continuity. This manuscript is suitable for publication in Hydrology, but requires major revision.
Abstract:
The Abstract contains significant gaps. As written, it fails to convey a clear overview of the paper’s objectives, scope, methods, and findings. A reader cannot grasp the study’s essence from the current text. Summarizing the entire paper in nine lines is insufficient; the Abstract should be substantially longer and more comprehensive, explicitly stating the aim, data sources, methodological approach, key results, and their implications.
Introduction:
The Introduction offers a broadly inclusive literature review, covering both Tibetan-Plateau studies and global analyses. However, most cited works are over six years old, and the review includes only a handful of articles published in the last five years. More recent research on surface water dynamics and water-level changes is available (, 2025 https://doi.org/10.1080/17538947.2025.2472924, 2025 https://doi.org/10.1016/j.jhydrol.2025.133948). These examples merely scratch the surface—many authors have published extensively on these topics. Please incorporate up-to-date references. Additionally, the Introduction does not sufficiently highlight the novelty of this work. What gaps did prior studies leave unaddressed, and how does this study fill them? Emphasize the original contributions, taking into account the latest literature.
Data and Methods:
This section is well written. All data sources and methods are clearly described—commendations to the authors for that clarity. My only criticism is that the figures appear low in resolution; please increase the DPI to ensure sharper images.
Results:
Organizing the Results into subheadings is logical and provides a clear presentation of the findings. Nevertheless, a few points remain unclear:
- What are the units of the statistical error metrics presented in Table 4?
- Are you certain about the y-axis label in Figure 14?
Moreover, the manuscript lacks a dedicated Discussion section. Very little of the Results is interpreted in depth. A scientific paper should include a thorough discussion that critically compares the present findings with those in the literature. Please add a Discussion section and transfer all text in the Results that belongs there into this new section.
Conclusion:
Where is the Conclusion section? It must not consist of only seven paragraphs. The Conclusion should summarize the study’s objectives, scope, limitations, and future recommendations. As one of the first sections a reader consults after the title and Abstract, it should provide a concise synthesis of the entire work, enabling readers to understand the paper’s structure and contributions at a glance.
In summary:
- The Abstract must be substantially expanded and rewritten to clearly state the study’s aim, scope, methods, and key findings.
- The Introduction should include up-to-date references (past five years) and explicitly highlight the paper’s novel contributions.
- Add a dedicated Discussion section to interpret and compare findings.
- A comprehensive Conclusion must be added, summarizing objectives, limitations, and future directions.
Reviewer 2 Report
Comments and Suggestions for Authors
This manuscript “Daily Water Mapping and Spatiotemporal Dynamics Analysis over the Tibetan Plateau” investigates lake dynamics on the Tibetan Plateau using 250 m MODIS daily reflectance time-series data (2000–2024) for daily water mapping. While the topic is relevant and potentially valuable, the manuscript suffers from issues in clarity, structure, and presentation. The main problems are unclear expression, excessive and unnecessary figures, non-standard figure formats, and insufficient temporal and driving factor analysis. Overall, the paper requires major revision and resubmission. Specific comments are as follows:
- Abstract
The abstract is overly brief. For example, the sentence “this study used 250 m MODIS daily reflectance time series data for daily water mapping from 2000 to 2024” should be improved by briefly describing the water extraction method employed.
The abstract should also summarize the key findings of the long-term water dynamics analysis, and ideally include one sentence highlighting the broader implications of the results for future studies or applications.
- Introduction
The introduction is currently weak. A strong introduction should:
Clearly identify a research problem or gap,
Provide a critical review of relevant literature,
Explicitly show how the current study addresses this gap.
At present, the research gap, scientific motivation, and contribution are not clearly articulated. Please revise to strengthen these aspects.
- Reference formatting
Line 58: The citation style for “Ji et al.” is incorrect. Please check and revise according to the journal’s reference format.
- Section 2. Data (line 79)
This section should describe the study area as well as the datasets. Adding information on the geographic location and climatic conditions of the Tibetan Plateau would strengthen the context.
Figure 1 is currently an uninformative data schematic. Normally, Figure 1 should present the study area map, including the location of the Tibetan Plateau within China, the distribution of lakes studied, and ideally the boundaries of the eight MODIS tiles used (“h23v05, h24v05, h24v06, h25v05, h25v06, h26v05, h26v06, and h27v06”).
- Figure 3
Line 136: The figure is difficult to interpret. For example, in panel (a), the legend values “0–100” appear to represent 0%–100%, but it is unclear how cloud cover in the southern mountainous regions of the Tibetan Plateau could reach 100%—does this imply year-round cloud cover?
In panels (b) and (c) (Namucuo Lake and Qinghai Lake), the lake boundaries are not visible, and their locations within the study area are unclear. Please clarify and improve the figure.
- Figure 11
Line 292: Please clarify whether the comparison with the JRC water product was performed on the same acquisition dates as the authors’ extracted lake areas. If not, this could bias the comparison.
- Figure 12
Line 293: The figure suffers from severe overlapping of graphical elements. Consider using a logarithmic scale to represent annual change rates, which may improve clarity.
- Figure 13 ()
Line 307: It is unclear why only these specific lakes were selected for individual area change analysis. Their locations should be indicated on the study area map.
9. Section 4. Result
Line 253: A brief analysis of the driving factors behind the observed lake area changes would greatly strengthen this section.
Reviewer 3 Report
Comments and Suggestions for Authors
dear authors,
the paper proposes daily surface water mapping over the entire Tibetan Plateau from 2000 to 2024, integrating MODIS Terra and Aqua data with different resolutions and spectral bands, and also including lake ice phenology.
The main innovative aspects are:
-Multi-source fusion of MOD09GQ/MYD09GQ (250m, 2 bands) and MOD09GA/MYD09GA (500m, 7 bands) to improve both spatial and spectral resolution.
Integration with LST data (MOD11A1) to improve ice/water classification.
-Advanced temporal interpolation and superpixel mapping algorithms to reduce cloud effects and improve accuracy.
-Detailed spatiotemporal analysis on 1293 lakes, with annual and seasonal trends.
These elements represent an interesting methodological contribution, although based on already known data and techniques.
The contribution is significant for the community studying water balance and climate impacts over Central Asia, although it does not introduce new theoretical paradigms.
Nevertheless some critical issues can be found:
1) Methodological
Empirical assumptions: some thresholds (e.g., T_LST, T_ICE) are based on “experience” and not on statistical analysis or literature.
LST resolution: ice classification is based on 1 km data, which may lose detail in smaller lakes.
Superpixel mapping: the KCCE method is interesting but little known; a more accessible explanation or comparison with more widely used methods would be needed.
2)Writing
The text is too technical and sectional, with dense formulas and descriptions that could be simplified for a wider audience.
Some sections (e.g., fusion rules) are complex and could benefit from diagrams or pseudocode.
3) Validation
Recall for ice is only ~81%, indicating underestimation of lake ice.
The accuracy for small lakes is lower, as admitted by the authors.
It is therefore suggested that you improve the paper on these aspects, and in particular to:
-clarify the empirical thresholds;
-improve the readability of some sections;
-discuss better the limitations related to LST resolution and ice classification.
Round 2
Reviewer 1 Report
Comments and Suggestions for Authors
I previously reviewed this manuscript. The authors have provided responses and made additions, and the manuscript has generally improved. However, there are still important shortcomings in the Discussion section. In particular, there is no deep discussion that compares the results with key studies in the literature. Lake surface area change is a widely studied and important topic; therefore, the Discussion should critically situate this study’s findings within the existing literature (such as 10.29128/geomatik.1017376, 10.1016/j.jhydrol.2025.133645).
Moreover, the use of MODIS data with low spatial resolution must be addressed explicitly. How does the low spatial resolution of MODIS affect the study’s results? How accurate is shoreline detection when using MODIS? These issues and questions should be discussed in detail in the Discussion.
Conditional on strengthening the Discussion as noted above, the manuscript can be considered publishable. My recommendation is minor revision.
Reviewer 2 Report
Comments and Suggestions for Authors
The manuscript, “Daily Water Mapping and Spatiotemporal Dynamics Analysis over the Tibetan Plateau,” has been thoroughly revised in response to the previous review comments. Overall, the authors have made substantial improvements, and the manuscript is now suitable for publication in Hydrology. Nevertheless, a few minor issues should be addressed before final acceptance:
Line 11: The unit “km2” is not written in the standard format and should be corrected (e.g., “km²”).
Line 264 (Figure 7): The figure is not visually clear. Please redraw it, ensuring that: the first two subplots in the first row are of equal size, the upper and lower rows are properly aligned, and the legend is placed in a clearer and more consistent position.
